# Low Molecular Weight Fucoidan Prevents Radiation-Induced Fibrosis and Secondary Tumors in a Zebrafish Model

**DOI:** 10.3390/cancers12061608

**Published:** 2020-06-18

**Authors:** Szu-Yuan Wu, Wan-Yu Yang, Chun-Chia Cheng, Ming-Chen Hsiao, Shin-Lin Tsai, Hua-Kuo Lin, Kuan-Hao Lin, Chiou-Hwa Yuh

**Affiliations:** 1Department of Food Nutrition and Health Biotechnology, College of Medical and Health Science, Asia University, Taichung 42354, Taiwan; szuyuanwu5399@gmail.com; 2Division of Radiation Oncology, Department of Medicine, Lo-Hsu Medical Foundation, Lotung Poh-Ai Hospital, Yilan 265, Taiwan; 3Big Data Center, Lo-Hsu Medical Foundation, Lotung Poh-Ai Hospital, Yilan 265, Taiwan; 4Department of Healthcare Administration, College of Medical and Health Science, Asia University, Taichung 41354, Taiwan; 5School of Dentistry, College of Oral Medicine, Taipei Medical University, Taipei 110, Taiwan; 6Institute of Molecular and Genomic Medicine, National Health Research Institutes, Zhunan, Miaoli 35053, Taiwan; cs081011@nhri.edu.tw (W.-Y.Y.); cccheng.biocompare@gmail.com (C.-C.C.); aenny716@gmail.com (S.-L.T.); hklin66@gmail.com (H.-K.L.); khlin@nhri.edu.tw (K.-H.L.); 7Radiation Biology Research Center, Institute for Radiological Research, Chang Gung University/Chang Gung Memorial Hospital at Linkou, Taoyuan 33302, Taiwan; 8Research and Development Center, Hi-Q Marine Biotech International Ltd., Songshan District, Taipei 10561, Taiwan; ming.hsiao@hiqbio.com; 9Institute of Bioinformatics and Structural Biology, National Tsing-Hua University, Hsinchu 30013, Taiwan; 10Department of Biological Science & Technology, National Chiao Tung University, Hsinchu 30010, Taiwan; 11Program in Environmental and Occupational Medicine, Kaohsiung Medical University, Kaohsiung 80708, Taiwan

**Keywords:** Oligo-Fucoidan, zebrafish, radiation-induced fibrosis, radiation-induced secondary malignancy

## Abstract

Radiotherapy often causes unwanted side effects such as radiation-induced fibrosis and second malignancies. Fucoidan, a sulfated polysaccharide extracted from brown seaweed, has many biological effects including anti-inflammation and anti-tumor. In the present study, we investigated the radioprotective effect of Oligo-Fucoidan (OF) using a zebrafish animal model. Adult zebrafish of wild-type and transgenic fish with hepatocellular carcinoma were orally fed with Oligo-Fucoidan before irradiation. Quantitative PCR, Sirius red stain, hematoxylin, and eosin stain were used for molecular and pathological analysis. Whole genomic microarrays were used to discover the global program of gene expression after Oligo-Fucoidan treatment and identified distinct classes of up- and downregulated genes/pathways during this process. Using Oligo-Fucoidan oral gavage in adult wild-type zebrafish, we found Oligo-Fucoidan pretreatment decreased irradiation-induced fibrosis in hepatocyte. Using hepatitis B virus X antigen (HBx), Src and HBx, Src, p53−/+ transgenic zebrafish liver cancer model, we found that Oligo-Fucoidan pretreatment before irradiation could lower the expression of lipogenic factors and enzymes, fibrosis, and cell cycle/proliferation markers, which eventually reduced formation of liver cancer compared to irradiation alone. Gene ontology analysis revealed that Oligo-Fucoidan pretreatment increased the expression of genes involved in oxidoreductase activity in zebrafish irradiation. Oligo-Fucoidan also decreased the expression of genes involved in transferase activity in wild-type fish without irradiation (WT), nuclear outer membrane-endoplasmic reticulum membrane network, and non-homologous end-joining (NHEJ) in hepatocellular carcinoma (HCC) transgenic fish. Rescue of those genes can prevent liver cancer formation. Conclusions: Our results provide evidence for the ability of Oligo-Fucoidan to prevent radiation-induced fibrosis and second malignancies in zebrafish.

## 1. Introduction

Radiotherapy (RT) consists of high-energy X-ray irradiation that is meant to target rapidly dividing cells such as tumor cells. RT damages the DNA within cells, often resulting in impaired cell division and subsequent cell death [1,2]. However, RT entails a balance between destroying cancerous cells and minimizing damage to normal cells [3] since RT can kill dividing cancerous cells as well as dividing non-cancerous cells, resulting in many undesired side effects [3].

RT-induced secondary malignancies (SM) may develop after X-ray irradiation [4]. RT leads to DNA mutations and cell death directly and also generates free radical damage to essential cellular enzymes [4]. The severity of the effects of RT can be seen in radiation-induced liver disease (RILD) in which hepatic toxicity progresses to fibrosis, cirrhosis, and liver failure and consequently becomes fatal liver [5]. The secondary malignancy (SM) or so-called late effect may not be observed right after the end of treatment [6,7,8].

While the induction of SM by RT is the side effect of greatest concern, other potential common side effects of RT occur, which are mainly caused by tissue fibrosis [9,10]. One possible means of reducing the side effects of RT is simply reducing the radiation dosage levels of RT. However, this can result in incomplete damage to cancer cells and even stimulates cancer metastasis [11]. A number of radioprotectants, including amifostine, dexrazoxane, and mesna (sodium 2-mercaptoethane sulfonate), have been reported. New potential radioprotectants also have been developed including amifostine analog S-[2-(3-methylaminopropyl) aminoethyl] phosphorothioate acid, thiolamine compounds with thioglycoside-protecting groups, covalent conjugates of thioamines and antioxidant vitamins, and selenazolidine prodrugs [12]. Among them, amifostine exhibits potential preventing radiation-induced cell death and mutagenesis and has entered phase I clinical study [13], nicaraven prevents radiation-induced lung cancer metastasis [14], zoledronate prevents radiation-induced bone loss [15], and tangeretin inhibits irradiation-induced lung metastasis [16]. Nevertheless, none of them have studied the protective effect against irradiation-induced liver fibrosis.

We investigated alternative approaches to reducing the side effects of RT by Fucoidan. It is a sulfated polysaccharide extracted from brown seaweed exhibiting anti-inflammatory and anti-tumor activities [17]. Fucoidan exhibits anti-bacterial, anti-fungal, and anti-viral properties [18]. Fucoidan inhibited CCl4-induced liver fibrosis [19] and attenuated N-nitrosodiethylamine-induced liver fibrosis [20] via the tissue growth factor (TGF)-beta1/Small mothers against decapentaplegic (Smad) [21]. Fucoidan also has a protective effect against gamma-ray-induced blood cell damage [22]. The low molecular weight fucoidan (LMWF) was extracted from brown seaweed using enzyme hydrolysis to an average molecular weight of 800 Da and has more biological actions than native fucoidan, such as growth inhibition of a broad spectrum of human carcinoma cells, including cervix adenocarcinoma, fibrosarcoma, leukemia, lymphoma, lung adenocarcinoma, and leukemia cells (HeLa, HT1080, K562, U937, A549, HL-60); angiogenesis and invasion inhibition in fibrosarcoma cells (HT1080); induced apoptosis in breast cancer cell lines (MCF-7, MDA-MB-231); promoted anti-coagulant activity in fibroblast cell line (CCL39); anti-inflammatory property by decreased leukocyte accumulation and connective tissue proteolysis; and protect extra-cellular matrix, etc. [18]. In mouse model, fucoidan exhibits dose-dependent increases in hematopoietic cells number through antioxidation mechanisms [23]. However, none of them have investigated that the fucoidan reduced liver fibrosis and liver cancer induced by irradiation. In this study, we used adult zebrafish animal model for the Oligo-Fucoidan radioprotection effect.

The zebrafish are externally fertilized, development occurs externally, with short sex-maturation time and low maintenance costs, and their genome exhibits 87% homology to the human genome [24,25]. Therefore, zebrafish have become a popular and common model for human diseases [26,27]. We have established many zebrafish cancer models using a transgenic approach [28,29,30,31,32,33,34,35] and screened anti-cancer therapeutic means [36,37]. Recently, zebrafish have been used in radiation protection research [38,39]. Zebrafish embryos were used as a model to screen radiation modifiers [40]; ionizing radiation ranging from 10 to 40 Gy caused time- and dose-dependent perturbations and lethality. Adult zebrafish irradiated with 20 Gy can impact the outcome of hematopoietic cell transplant [41]. However, there is no literature for using adult zebrafish for radio-protection. Therefore, the dosage was determined based on a previous study in rat that fucoidan from *Laminaria japonica* at 300 mg/kg body weight per day has no adverse effect [42]. Herein, zebrafish adult fish were feeding with Oligo-Fucoidan (OF) by oral gavage; the dose was 300 mg/kg (0.051 mg/fish).

Previously, we demonstrated that liver-specific expression of hepatitis B virus X antigen (HBx) and *src* oncogene are highly associated with liver cancer [29]. Obesity is considered a significant risk factor for liver cancer and, using diet-induced-obesity (DIO), the developed hepatocellular carcinoma (HCC) can be facilitated with a higher incidence rate [31]. In this study, we used WT fish and HBx, src, p53-DIO transgenic zebrafish to investigate the effects of Oligo-Fucoidan on preventing radiation-induced fibrosis and secondary tumors.

## 2. Results

### 2.1. Oligo-Fucoidan Treatment Reduces the Expression of Apoptotic Genes, Prevents Radiation-Induced Fibrosis and Cell Proliferation Markers in Adult WT Zebrafish

To observe the radioprotective effects in adult zebrafish, Oligo-Fucoidan was orally administered to five-month-old wild-type fish thrice a week for one month and one week after irradiation. Liver tissue was then excised for qPCR analysis. Activation of caspases are essential for apoptosis, including initiators (caspase-2, -8, -9, -10), effectors (caspase-3, -6, -7), and inflammatory caspases (caspase-1, -4, -5). The intrinsic signaling pathways of apoptosis are regulated by B-cell lymphoma 2 (BCL2) family of proteins, among them, Bad (BCL2 antagonist of cell death) and Bax (BCL2-associated X protein) are pro-apoptotic proteins. Fas ligand (FasL) is initial step for extrinsic signaling pathways of apoptosis [43]. c-Jun N-terminal kinase (JNK) and p38 mitogen-activated protein kinase (MAPK) are key regulators of cell stress and often associated with apoptosis [44]. Therefore, we choose to examine the expression levels for *caspase8*, *caspase3a*, *bad*, *bax*, *fasL*, *jnk-1,* and *p38a* (*MAPK*) as markers for cell death. The expression of cell death/stress-related genes was examined by qPCR. After RT, the expression of cell death/stress-related genes was increased relative to non-irradiated wild-type fish (Appendix A). Pretreatment with Oligo-Fucoidan reversed the expression of cell death/stress genes induced by RT (Figure 1A). These results indicated that Oligo-Fucoidan can prevent high-dosage irradiation-induced liver cell apoptosis.

Type I collagen is associated with hepatic fibrosis [45]. Type I procollagen is combined with two pro-alpha1 (I) and one pro-alpha2 (I) chain, which are encoded by COL1A1 and COL1A2, respectively [46]. Connective tissue growth factor (CTGF) is a fibrogenic master switch in liver fibrosis [47]. Heparanase (HPSE) expression increased in the onset of liver fibrosis of CCl4-treated mice model [48]. Therefore, we chose to examine the expression levels for *col1a1*, *ctgfa*, and *hpse* as markers for liver fibrosis. The hepatic expression of the fibrotic marker genes was examined both in non-irradiated and irradiated WT fish. The expression of *col1a1* and *ctgfa* was increased in the liver after RT relative to non-irradiated fish, but the increment of *hpse* was not significant (Appendix A). Pretreatment with Oligo-Fucoidan reversed fibrotic marker induction following irradiation (Figure 1B).

The initiation of hepatocellular carcinoma depends on E-type cyclins E1 (CcnE1) and cyclin-dependent kinase 2 (Cdk2) [49]. Ccne1 overexpression cause liver tumor development in mice [50], Cdk2 plays a key role in cell cycle progression in hepatocyte [51], and cyclin-dependent kinase 1 (Cdk1) is essential for cell division of liver cancer [52]. Therefore, we chose to examine the expression levels for *ccne1*, *cdk1*, and *cdk2* as markers for cell proliferation. To evaluate the effect of Oligo-Fucoidan on hepatocarcinogenesis, the expression of cell cycle/proliferation markers were examined in WT fish. The expression of cell cycle/proliferation markers was increased in the liver after radiation treatment (Appendix A). Pretreatment with Oligo-Fucoidan reversed the expression of cell cycle/proliferation markers induced by radiation, especially the expression of *cdk1* and *cdk2* (Figure 1C). These results indicated Oligo-Fucoidan can prevent high-dosage irradiation-induced liver cell proliferation.

We wonder whether lower irradiation can have less effect on fibrosis, we irradiated the wild-type fish with 10 Gy, a dosage used for radiotherapy in human patients. It caused an increment of *ctgfa,* which might represent an early fibrosis marker (Appendix A). Again, pretreatment with Oligo-Fucoidan reversed fibrotic marker induction following irradiation (Figure 1D), but there was no statistical significance.

To verify the liver fibrosis caused by irradiation, Sirius red staining was used for visualization of collagen I and III fibers in liver sections. The 40 Gy irradiation-induced fibrosis and Oligo-Fucoidan pretreatment indeed reduced the formation of collagen fibers derived from radiation (Figure 2A,B). Irradiation with 10 Gy also caused liver fibrosis, which was prevented by Oligo-Fucoidan pretreatment (Figure 2C,D). These results indicated Oligo-Fucoidan can prevent a high dosage of irradiation-induced liver fibrosis.

### 2.2. Oligo-Fucoidan Pretreatment Decreases the Expression of Lipogenic Factors, Lipogenic Enzymes, Fibrosis, and Cell Proliferation Markers in Adult Transgenic Zebrafish

HBx,src transgenic zebrafish, in which the hepatitis B viral X antigen (HBx) and *src* oncogene are expressed in the liver, developed hepatocellular carcinoma (HCC) at 9–11 months of age [29]. Diet-induced obesity (DIO) accelerated HCC formation in HBx, src transgenic fish at 5 months of age when compared to normal diet, 23% of HCC under DIO versus 7% of HCC in normal diet [31]. To examine the effects of Oligo-Fucoidan on obesity-induced hepatocarcinogenesis, diet-induced obese HBx,src transgenic fish at 5-month of age were examined for the expression of lipogenic factor, lipogenic enzyme, and fibrosis markers relative to untreated fish.

The expression patterns of the lipogenic factors *pparg*, *srebf1,* and *mlxip* were examined (Figure 3A and Appendix A) as well as the expression patterns of the lipogenic enzymes *agpat4*, *pap,* and *fasn* (Figure 3B and Appendix A). These results demonstrate that pretreatment of fish with Oligo-Fucoidan can reduce the expression of lipogenic factors and lipogenic enzymes induced by the combination of diet-induced obesity and RT. Oligo-Fucoidan pretreatment significantly decreased the expression level of fibrotic marker genes following RT relative to RT alone (Figure 3C and Appendix A).

In the diet-induced obese HBx, src transgenic fish, RT increased the expression of cell cycle-related genes without statistical significance (Appendix A), and Oligo-Fucoidan dramatically reversed the expression of cell cycle/proliferation markers induced by radiation (Figure 3D). Ten grays, a dosage used for radiotherapy in human patients, did not cause increment of fibrosis markers (Appendix A) but lowered the expression of cell proliferation markers without significance (Appendix A). Oligo-Fucoidan pretreatment significantly decreased the expression levels of fibrotic marker *ctgfa* (Figure 3E) and cell proliferation marker *ccne1* (Figure 3F).

While RT is a method of treating tumors, radiation can also induce secondary tumors [53]. In our experiments, we found that the expression of lipogenic factors, fibrosis, and cell proliferation markers was either unchanged or increased following 40 Gy RT, indicating that the dosages of 40 Gy RT may be too high and may facilitate secondary malignancy. The expression of lipogenic factors and fibrosis and cell proliferation markers in the RT group could be reversed by pretreatment with oligo-fucoidan. These results provide strong evidences for the radioprotective effect of Oligo-Fucoidan. 

### 2.3. Oligo-Fucoidan Pretreatment Decreases the Radiation-Induced Hepatocyte Apoptosis in Adult Zebrafish

RT-induced cell death was mediated through double-stranded DNA sensor AIM2, which senses radiation-induced DNA damage in the nucleus and then activates inflammation and cell death [54]. RT-induced cell death depends on DNA repair capacity and the microenvironment on top of other factors [55]. Hepatocellular apoptosis may trigger fibrosis and tumor formation [56]. We detected hepatocyte apoptosis from hematoxylin and eosin (H&E) stain liver specimens to verify the apoptosis changes following Oligo-Fucoidan treatment in transgenic fish. We found in our zebrafish model that radiation indeed induced apoptosis in the liver tissue from H&E stain, and Oligo-Fucoidan-fed zebrafish significantly reduced the hepatocyte apoptosis (Figure 4).

### 2.4. Oligo-Fucoidan Pretreatment in Adult Transgenic Zebrafish Decreased HCC Formation

We then examined liver specimens using H&E staining to verify the histopathological changes following Oligo-Fucoidan treatment in transgenic fish. When overfed, HBx, src transgenic zebrafish (DIO) exhibited hyperplasia (80%) and HCC (20%). Treatment with 40 Gy radiation (DIO + 40 Gy) increased the HCC risk from 20% to 44% (Figure 5A). Oral feeding of Oligo-Fucoidan before radiation (DIO + OF + 40 Gy) significantly decreased the HCC incidence from 44% to 10% (Figure 5A). These data indicate that Oligo-Fucoidan effectively decreases the risk of radiation-induced HCC. Ten grays irradiation decreased the HCC incidence (Figure 5B), and Oligo-Fucoidan treatment further reduced the percentage of dysplasia (Figure 5B).

The proliferating cell nuclear antigen (PCNA) proliferation marker was also examined in liver specimens using immunohistochemical staining in two transgenic fish with irradiation. The PCNA staining was increased in the radiation and DIO groups and was reduced by Oligo-Fucoidan pretreatment (Figure 5C); however, 10 Gy of irradiation did not cause a significant increase of PCNA, hence Oligo-Fucoidan had no reversal effect (Figure 5D). These data suggest that Oligo-Fucoidan is not only radioprotective but it can also decrease the expression of proliferation markers, thus lowering the possibility of cancer formation and secondary malignancies induced by radiation.

### 2.5. GeneTitan Array Analysis Identified Important Targets for Oligo-Fucoidan Radioprotective Effect

In order to explore global program of gene expression after Oligo-Fucoidan treatment before irradiation and identified distinct classes of up- and downregulated genes/pathways during this process in hepatocytes, the whole-genome expression profiles were analyzed by GeneTitan array. We compared two batches of wild-type fish radiation (1st batch-40 Gy and 2nd batch-10 Gy) with or without Oligo-Fucoidan pretreatment. We identified 626 candidate genes that were downregulated by irradiation (R/control) and upregulated by Oligo-Fucoidan pretreatment before radiation (OF + R/R) at least two-fold with a *p*-value less than 0.05 (Figure 6A,B). Using gene ontology analysis via WebGestalt [57], we found genes involved in oxidoreductase activity were enriched (Figure 6C). All the differentially expressed genes were uploaded to the NetworkAnalyst 3.0 software (https://www.networkanalyst.ca/). hnf4a was the hub for most of the differentially expressed genes. Therefore, *hnf4a* was determined to be a driver gene. It was reported that stress-induced inhibition of HNF4A expression is related to liver fibrosis and cancer formation [58]. Overexpressed Hnf5a resets the transcription factor network in hepatocytes and prevents hepatic failure [59]. Therefore, we examined fucoidan-mediated activation of hnf4a in the following section.

We also identified 149 candidate genes that were upregulated by irradiation (R/control) and downregulated by Oligo-Fucoidan pretreatment before radiation (OF + R/R) at least two-fold with a *p*-value less than 0.05 (Figure 6D,E); these genes were enriched in transferase activity (Figure 6F).

We compared two batches of transgenic fish radiation (1st batch-HBx, src fish diet-induced obesity (DIO) with 40 Gy and 2nd batch- HBx, src, p53-transgenic fish DIO with 10 Gy) with or without Oligo-Fucoidan pretreatment. We identified 502 genes that were downregulated by 40 Gy irradiation in HBx,src transgenic fish with DIO (DIO + R/DIO) and upregulated by Oligo-Fucoidan pretreatment before radiation in both batches (DIO + OF + R/R) at least two-fold with a *p*-value less than 0.05, and these genes are involved in oxidoreductase activity (Figure 7A–C).

We also identified 234 genes that were upregulated by 40 Gy irradiation in HBx,src transgenic fish with DIO (DIO + R/DIO) and upregulated by Oligo-Fucoidan pretreatment before radiation in both batches (DIO + OF + R/R) at least two-fold with a *p*-value less than 0.05, and those genes involved Oligo-Fucoidan also decreased the expression of genes involved in nuclear outer membrane-endoplasmic reticulum membrane network and non-homologous end-joining (NHEJ) in HCC transgenic fish (Figure 7D–F). Rescue of those genes can prevent liver cancer formation.

### 2.6. Enhancement of Hnf4a, Asgpr, and Hnf4a Downstream Target Genes by Oligo-Fucoidan Pretreatment in X-Irradiated WT and HBx, Src, p53−/+ Transgenic Zebrafish 

In order to explore fucoidan-mediated gene expression in hepatocytes, the whole-genome expression profiles of four groups of fish (wild-type (WT), WT with irradiation, HCC, and HCC with irradiation) treated with Oligo-Fucoidan were analyzed by GeneTitan array, and 23 candidate genes displaying statistically significant differential expression were filtered out (Figure 8A). HNF4A was determined to be a driver gene after analysis by NetworkAnalyst 3.0 (Figure 8B).

Since we identified *hnf4a* is the driver gene of Oligo-Fucoidan upregulated genes from WT and HCC irradiation, we verified more fish with qPCR. Although irradiation decreased or unchanged *hnf4a* expression (Appendix A), Oligo-Fucoidan pretreatment before irradiation in WT fish increased the *hnf4a* expression significantly (Figure 9A). We also verified the expression of dimethylarginine dimethylaminohydrolase 1 (*ddah1*) *and* 11Beta-hydroxysteroid dehydrogenase type 1 (*11b-hsd1*) under the radiation (Appendix A). *ddah1* increases nitric oxide (NO) synthase activity by degradation of the Nitrous Oxide Systems (NOS) inhibitor (ADMA), it has been reported to have a protective effect against high-fat-diet-induced fatty liver [60]. *11b-HSD1* catalyzes the regeneration of active cortisol from inert cortisone and was also upregulated by Oligo-Fucoidan before irradiation (Figure 9A). Those genes induced by fucoidan were considered to be associated with radioprotection effects, because NO and active cortisol have anti-inflammation, antitumor, and fibrosis-prevention activity.

Oligo-Fucoidan might bind to C-type lectin-like receptor 2 (CLEC-2) [61], and asialoglycoprotein receptor (ASGPR) is hepatocyte-specific CLEC-2. To verify whether Oligo-Fucoidan can affect the expression of ASGPR orthologues-zhi (asgpr1) and hnf4a downstream target gene-tdo2a, qPCR analysis was applied to those irradiation WT fish with or without Oligo-Fucoidan pretreatment. We found that Oligo-Fucoidan pretreatment induced ASGPR orthologues-*zhi* (*asgpr1*), and *tdo2a* expressions after 10 Gy X-irradiation (Figure 9B). The *zhi (asgpr1)* and *tdo2a* expressed with positive correlations (Figure 9C). Highly consistent data were also obtained in the *HBx*, *src*, *p53-/+*fish with DIO and Oligo-Fucoidan pretreatment (Figure 9D–F). Our preliminary data has demonstrated that Oligo-Fucoidan binds to ASGPR1/2 via in-vitro and in-vivo competition assay and increases the expression of *HNF4A* through the JAK2/STAT3 axis in hepatocyte.

## 3. Discussion

Fucoidan has been demonstrated to have several benefits, including anti-inflammatory and tumor retardation effects [62,63,64]. The Oligo-Fucoidan used in this study has a low molecular weight (500~1500 Da.), which showed greater functional effects and bioactivity compared to high molecular weight fucoidan [65]. In this study, we provided evidence that irradiation with either 40 Gy or 10 Gy promoted fibrosis in WT fish. We also used HCC zebrafish model irradiated with 40 Gy or 10 Gy and found higher irradiated dose caused cell proliferation and 10 Gy lower the expression of proliferation markers. Nevertheless, we successfully demonstrated the radioprotective effect of Oligo-Fucoidan on fibrosis, lipid accumulation, and cell cycle/proliferation by RT in zebrafish models.

In mice, oral gavage with Oligo-Fucoidan for 14 days can attenuate RT-induced pneumonitis and fibrosis in lung tissue specimens; the radiation dosage was 10 Gy/shot [66]. Fucoidan exhibits radioprotection effect against gamma-radiation-induced damage of blood cells [22]. In this study, we found the dosage of radiation at 40 Gy/shot or 10 Gy/shot was sub-lethal in zebrafish assessed by fish mortality, this result is similar to the previous study using adult zebrafish [41]. However, we revealed 40 Gy of irradiation not only cannot reduce cancer formation but also induced severe fibrosis and HCC formation. We observed that RT induced higher incidences of cancers, which was reported previously [53,67,68,69]. Reduced radiation dosage to 10 Gy/shot, which is similar to human patient received accumulatively in clinical treatment, revealed minor reduction of HCC in transgenic fish, but still induced fibrosis. HCC developed in DIO HBx, src transgenic fish with 40 Gy irradiation can be prevented by oligo-fucoidan, which indicates the anti-tumor effect of fucoidan. 

A number of radio-protectants have been reported, amifostine ameliorated radiation-induced mutagenesis but is limited by its inherent systemic toxicity [70], nicaraven had very limited effects on tumor growth [71], and although tangeretin reduces irradiation-induced lung metastasis, it provokes food-drug interactions via causing CYP3A4 induction [72]. In this study, we used Oligo-Fucoidan extracted from brown seaweed by enzymatic hydrolysis to lower molecules at the average molecular weight of 800 Da, which has been demonstrated no toxicological up to 2000 mg/kg BW/day and is a safe food supplement [73]. The current study is, if any, the first article to demonstrate an effective agent in preventing radiation-induced secondary cancers. For radiation oncologists and their patients, this may be promising liver cancer or treatment of secondary malignancies. Further clinical trials will be valuable.

The structure and molecular weight of fucoidan as well as the methods to generate the low molecular weight fucoidan affect the bioactivity greatly. Low molecular weight fucoidan had significantly higher capacity [74], but acidic hydrolysis is difficult to obtain the proper size due to the cleavage properties [75]. Fucoidans are sulfated l-fucose (6-deoxy-l-galactose) polysaccharide in general, but their structures vary dramatically according to species, season, location, and maturity [76]. The Oligo-Fucoidan was extracted from *Laminaria japonica*, and enzymatic cleavage to an average molecular weight in a range between 500 and 1500 Da. The structural and chemical property of fucoidan from *Laminaria japonica* has been reported and the low molecular weight was more potent than the medium molecular weight ones [77].

Fucoidan exhibits numerous bioactivities including anti-inflammation, antioxidant, anti-coagulant, and anti-cancer through many different signaling pathways [78]. Mechanically, Oligo-Fucoidan has been reported to suppress activation of MAPK/PI3K pathway on triple-negative breast cancer (TNBC) [79], and inhibitions of MAPK/PI3K is related to increase of apoptosis genes in cancer cells. Fucoidan exhibits anticancer effect via increased Bax/Bcl-2 expression ratio in cancer cells [80]. However, we found fucoidan decreased radiation-induced increments of Bad, bax, and other apoptosis genes’ expression. Therefore, the function of fucoidan in irradiated cells versus cancer cells seems to be opposite, while Fucoidan exhibits cytoprotection effect, it also inhibits cancer cells. Oligo-Fucoidan also inhibits migration of TNBC cells in zebrafish xenotransplantation model [79]. In our other projects, the preliminary results showed that Oligo-Fucoidan promotes the cell viability of normal liver cells while inhibits the cell growth for multiple hepatoma cells.

The inhibition of collagen I accumulation of Oligo-Fucoidan’s radioprotection was mediated through TGF-beta1/Smad pathway [81]. Using whole genomic microarray to screen the genes are down- or upregulated by irradiation and reversed by Oligo-Fucoidan pretreatment before irradiation is highly related to the clinical setting, and we identified genes are involved in radiation and reduced by Oligo-Fucoidan pretreatment proving the efficacy of Oligo-Fucoidan on radioprotection. In this study, we identified 626 candidate genes enriched in oxidoreductase activity were downregulated by irradiation in WT and reverted by Oligo-Fucoidan pretreatment before irradiation. We identified 502 genes enriched in oxidoreductase activity were downregulated by 40 Gy irradiation in HBx, src transgenic fish and reverted by Oligo-Fucoidan pretreatment before irradiation. The genes involved in oxidoreductase activity were upregulated by Oligo-Fucoidan pretreatment in both WT and HCC transgenic fish, which reveal their importance. It was reported that the expression of xanthine dehydrogenase (XDH) was decreased by radiation in liver of mice [82]. Reactive oxygen species (ROS) produced during irradiation to kill tumor cells, and these cells were also involved in the development of radiation-induced normal tissue damage (RINTD). Oxidative stress induced by irradiation [83] caused liver inflammation, fibrosis, and apoptosis [84]. Gene expression profiling revealed radiation changes metabolic pathways responsible for adaptation and maintenance of homeostasis in mice [85]. Our genomic profile studies reveal future, potential genes of interest for radioprotection.

Our transcriptomic analysis data also demonstrated that genes involved in nuclear outer membrane-endoplasmic reticulum membrane network and non-homologous end-joining (NHEJ) were upregulated by 40 Gy irradiation in HBx, src transgenic fish with DIO and downregulated by Oligo-Fucoidan pretreatment. Radiation-induced perturbation organization of different cell organelle membranes is highly related to acute radiation injury [86]. Ionizing radiation also induced DNA double-strand breaks (DSBs) [87], which can lead to cancers. The genes related to NHEJ were revealed by microarray as radiation-responsive biomarkers [88]. In our study, we found Oligo-Fucoidan treatment before irradiation in HCC transgenic fish can decrease the expression of genes involved in nuclear outer membrane-endoplasmic reticulum membrane network and NHEJ provide hints for the radioprotection effect of Oligo-Fucoidan in HCC. 

From all the differentially expressed genes analyzed by NetworkAnalyst, hnf4a was the hub for most of the differentially expressed genes. Therefore, hnf4a was determined to be a driver gene. Fucoidan was reported to act through many different signaling pathways [78], but the central target gene and its receptor were unknown. In our other project, we demonstrated Oligo-Fucoidan binds to ASGPR1/2 receptor in normal hepatocyte or hepatoma cells through both in-vitro and in-vivo competition assay, and activate pSTAT3 and bind to the promoter of HNF4a-P1 isoform and increase the expression of Hnf4a (unpublished result). Hnf4a is an important transcription factor in mouse liver, and increased expression of Hnf4a can prevent hepatic failure [59]. We found that Oligo-Fucoidan treatment before irradiation significantly increased the expression of hnf4a in zebrafish model. We further found the expression of *ddah1* and *11b-hsd1* was upregulated by Oligo-Fucoidan treatment before irradiation. The *ddah1* can increase nitric oxide (NO) synthase activity by degradation of the NOS inhibitor, ADMA, and was reported to have protective effects against a high-fat, diet-induced fatty liver [60], and *11beta-HSD1* catalyzes the regeneration of active cortisol from inert cortisone. Our study demonstrated that Oligo-Fucoidan comprehensively modulates multiple pathways and protects hepatocyte against irradiation-induced fibrosis and hepatocarcinogenesis in zebrafish. Interestingly, this mechanism of action is different from that of many other small, molecular drugs, and fucoidan represents a promising avenue for future research and clinical treatment.

## 4. Materials and Methods 

### 4.1. Overall Design

Forty wild-type (WT) 5-month-old adult fish were divided into two groups treated with γ-irradiation (R), and one group was fed with Oligo-Fucoidan for one month followed by 40 Gy γ-irradiation exposure (OF + R). Two batches of WT fish were performed, the first was treated with 40 Gy and the second was treated with 10 Gy. Three-month-old female HBx, Src and HBx, Src, p53–/+transgenic zebrafish were diet-induced obesity (DIO) for two months by approximately 6.9 mg and 83 mg cysts/fish/day respectively, and then 5-month-old adult fish were divided into two groups treated with γ-irradiation (R) and one group of fish oral feeding with Oligo-Fucoidan for one month followed by γ-irradiation exposure (OF + R). The γ-irradiation exposure for HBx, Src and HBx, Src, p53^–/+^ were 40 Gy and 10 Gy, respectively. Oligo-Fucoidan (300 mg/kg) was administrated by oral gavage for one month starting at 4 months of age for one month. The fish were sacrificed and the hepatic tissue was immediately frozen in liquid nitrogen and stored at −80 °C for later RNA extraction and hybridization on Affymetrix microarrays.

### 4.2. Oligo-Fucoidan (OF)

The low molecular weight fucoidan was provided by Hi-Q Marine Biotech International Ltd. (Taipei City, Taiwan). It was extracted from brown seaweed with enzyme hydrolysis to an average molecular weight in a range between 500 and 1500 Da.

### 4.3. Zebrafish Maintenance and Transgenic Zebrafish Lines

The Danio rerio AB strain, HBx,src transgenic fish: Tg (fabp10a: HBV-HBx-mCherry, myl7: EGFP) xTg (fabp10a:src, myl7:EGFP), and HBx, src, p53−/+ transgenic fish: Tg (fabp10a:HBV-HBx-mCherry, myl7:EGFP, p53−/+) xTg (fabp10a: src, myl7: EGFP, p53−/+) were used in this study. Danio rerio AB strain was obtained from the Zebrafish International Resource Center (ZIRC, Eugene, OR, USA), and the other transgenic lines were bred in our laboratory. Husbandry was followed by standard operation protocol. All of the fish were maintained in the Zebrafish Core Facility at our Institute (National Health Research Institutes, Miaoli, Taiwan). All experiments were approved by the Institution Animal Care and Use Committee (IACUC) of the National Health Research Institutes (NHRI-IACUC-106118-A).

### 4.4. Oligo-Fucoidan and Radiation Irradiation in Adult Zebrafish

The wild-type fish were fed normal diet, the amount of Artemia fed was approximately 6.9 mg cysts/fish/day. Three-month-old female HBx, Src and HBx, Src, p53–/+transgenic zebrafish were diet-induced obesity (DIO) fed with 12× of Artemia approximately 83 mg cysts/fish/day for two months. Oligo-Fucoidan (300 mg/kg) was administrated by oral gavage thrice a week for one month starting at 4 months of age for one month. The dosage was determined based on previous study in rats that fucoidan from L. japonica at 300 mg/kg body weight per day has no adverse effect [42]. The average weight of adult fish was 0.17 g, 300 mg/kg = 0.3 mg/g, 0.3 mg/g × 0.17 g = 0.051 mg. So, we diluted 0.051 mg in 5 µL of water and injected slowly into the intestinal tract of anesthetized (with 150 mg/L MS-222) adult fish using flexible tubing as described previously [89]. Then, fish were irradiated by 6MV X-ray of 40 Gy or 10 Gy at the dose rate of 500 Gy/min followed by oral gavage with Oligo-Fucoidan thrice, with a one-week interval. The fish were sacrificed and the hepatic tissue was immediately frozen in liquid nitrogen and stored at −80 °C for later RNA extraction and hybridization on Affymetrix microarrays.

### 4.5. Tissue Collection

At the end of the experiment, the fish were sacrificed and the tissue was removed and divided into two sections: one for RNA isolation and the other for histopathological analysis. For the RNA isolation, the tissues were immediately frozen in liquid nitrogen and stored at −80 °C for RNA isolation later. For histopathological analysis, the tissues were fixed in a 10% formalin solution (Sigma-Aldrich Inc., St. Louis, MO, USA), embedded in paraffin, sectioned at a thickness of 5 µm, mounted on Poly-L-lysine-coated slides, and stained with hematoxylin and eosin (H&E).

### 4.6. Total RNA Isolation

Total RNA was isolated by NucleoSpin^®^ RNA kit (Macherey-Nagel, Dueren, Germany) following the manufacturer manual, and RNA samples were eluted in 40 µL RNase-free water. The RNA samples were stored at −80 °C. For GeneTitan array analysis, RNA quantity and purity were assessed using NanoDrop ND-1000. Pass criteria for absorbance ratios were established as A260/A280 > 1.8 and A260/A230 > 1.5 average to be acceptable. RNA integrity RIN# was determined by Agilent 2100 Bioanalyzer with RNA 6000 Nano Assay with passing criteria set at >7.

### 4.7. Reverse Transcription Reaction (RT)

Complementary DNA (cDNA) was synthesized as described previously [29] by using an iScript^™^ cDNA Synthesis Kit (Bio-Rad, Hercules, CA, USA). The cDNA was stored at −20 °C for long-term preservation.

### 4.8. Quantitative Polymerase Chain Reaction (qPCR)

Real-time quantitative PCR was performed as described previously [29]. The specific primers used in the qPCR are listed in Appendix A. All experiments were performed in triplicate, and the mean values were obtained. At least three independent samples were used for the qPCR, and the medians and standard errors were calculated and are presented as the median ± standard error.

### 4.9. Hematoxylin and Eosin Staining and PCNA Immunohistochemistry Staining

The H&E staining and PCNA IHC staining were performed as described previously [29].

### 4.10. GeneTitan™ Array for Gene Expression Profiling

ZebGene 1.1 ST Array Plates (Affymetrix, Santa Clara, CA, USA) were used for the whole genomic transcriptome analysis. Transcriptome Analysis Console (TAC) software was used to analyze differentially expressed genes from Oligo-Fucoidan-treated versus control, non-treated fish. Expression Analysis Settings: Gene-Level Fold Change <−2 or >2, Gene-Level *p*-value < 0.05, Anova Method: ebayes. The raw data of the microarray have been submitted to the NCBI Gene Expression Omnibus (GEO) (http://www.ncbi.nlm.nih.gov/geo/) under accession code GSE148811. The gene ontology analysis was performed using gene ontology analysis via WEB-based GEne SeT AnaLysis Toolkit (WebGestalt [57], http://www.webgestalt.org/), pathway analyses were done by NetworkAnalyst 3.0 (http://www.networkanalyst.ca/), and activated pathways were selected and matched according to the Kyoto Encyclopedia of Genes and Genomes (KEGG) database.

### 4.11. Statistical Analysis

All data were analyzed using SPSS 17.0 (SPSS, Inc.). The statistical analysis was performed using two-tailed Student’s *t* tests. In all the statistical analyses, *p-*value < 0.05 were considered to be statistically significant and are presented as: *: 0.01 < *p* ≤ 0.05; **: 0.001 < *p* ≤ 0.01; and ***: *p* ≤ 0.001.

## 5. Conclusions

In conclusion, we proved Oligo-Fucoidan prevents radiation-induced fibrosis in zebrafish model. Oligo-fucoidan also prevents second malignancies induced by high dosage of radiation in zebrafish model. Using transcriptomic analysis, we found Oligo-Fucoidan pretreatment increased the expression of genes involved in oxidoreductase and decreased transferase and nuclear outer membrane-endoplasmic reticulum membrane network and non-homologous end-joining (NHEJ).

## Figures and Tables

**Figure 1 cancers-12-01608-f001:**
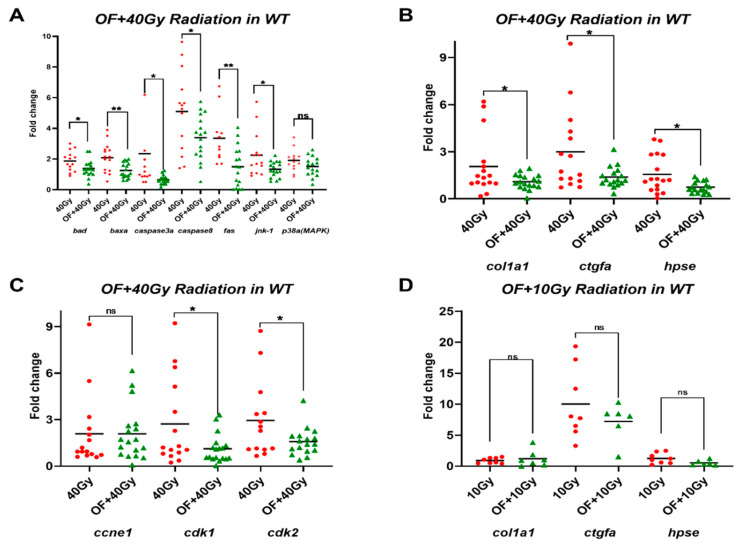
Expression levels of apoptosis, fibrosis, and cell proliferation-related genes in adult wild-type zebrafish treated with Oligo-Fucoidan and radiation. (**A**) The expression of B-cell lymphoma 2 (bcl2)-associated agonist of cell death (*bad*), bcl2-associated X protein a (*baxa*), apoptosis-related cysteine peptidase 3a (*caspase 3a*), apoptosis-related cysteine peptidase 8 (*caspase 8*), apoptosis antigen 1 (*fas*), c-Jun N-terminal kinase 1 (*jnk-1*), mitogen-activated protein kinase 8a (*p38(MAPK*)) in wild-type fish treated with 40 Gy irradiation (40 Gy) or treatment of Oligo-Fucoidan before 40 Gy irradiation (OF + 40Gy). (**B**) The expression of collagen, type I, alpha 1a (*col1a1*), connective tissue growth factor a (*ctgfa*), heparanase (*hpse*) in wild-type fish treated with 40 Gy irradiation (40 Gy) or treatment of Oligo-Fucoidan before 40 Gy irradiation (OF + 40 Gy). (**C**) The expression of cyclin E1 (*ccne1*), cyclin-dependent kinase 1 (*cdk1*), and cyclin-dependent kinase 2 (*cdk2*) in wild-type fish treated with 40 Gy irradiation (40 Gy) or treatment of Oligo-Fucoidan before 40 Gy irradiation (OF + 40 Gy). (**D**) The expression of *col1a1*, *ctgfa,* and *hpse* in wild-type fish treated with 10 Gy irradiation (10 Gy) or treatment of Oligo-Fucoidan before 10 Gy irradiation (OF + 10 Gy). The data are presented as dot plots with a horizontal line for the mean and are repeated in triplicate. The statistical significance was calculated using Student’s *t*-test (** p <* 0.05, *** p <* 0.01, ns: non-significant).

**Figure 2 cancers-12-01608-f002:**
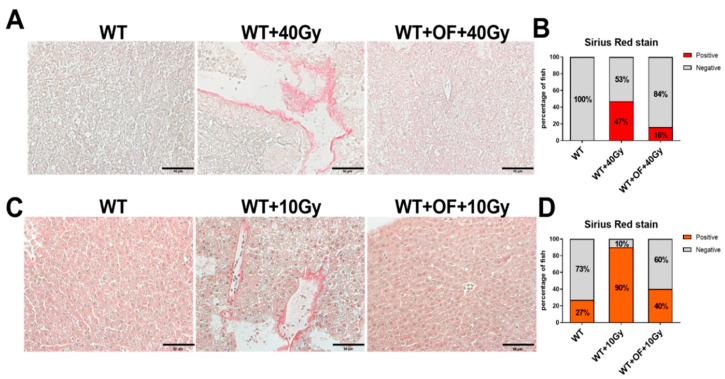
Sirius red staining in adult zebrafish treated with Oligo-Fucoidan and radiation. (**A**) The representative images of Sirius red staining of wild-type fish without irradiation (WT), with 40 Gy irradiation (WT + 40 Gy), or treatment of Oligo-Fucoidan before 40 Gy irradiation (WT + OF + 40 Gy). The images were taken at 400× magnification, and the scale shown is for 50 µm. (**B**) Statistical analysis from wild-type fish treated with 40 Gy irradiation. (**C**) The representative images of Sirius red staining of wild-type fish without irradiation (WT), with 10 Gy irradiation (WT + 10 Gy) or treatment of Oligo-Fucoidan before 10 Gy irradiation (WT + OF + 10 Gy). The images were taken at 400× magnification, and the scale shown is for 50 µm. (**D**) Statistical analysis from wild-type fish treated with 10 Gy irradiation.

**Figure 3 cancers-12-01608-f003:**
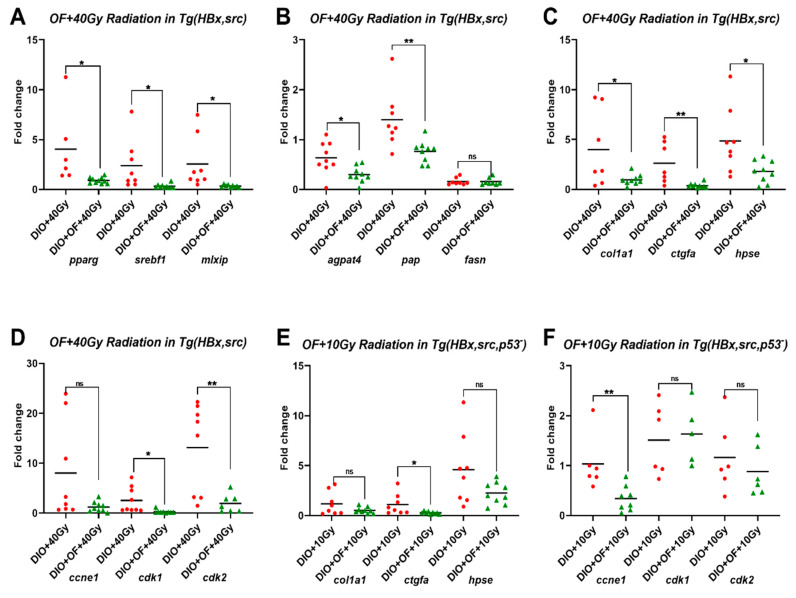
Expression levels of lipogenic factor and lipogenesis enzyme genes in adult zebrafish treated with Oligo-Fucoidan and radiation. (**A**) The expression of lipogenic factors (*pparg*, *srebp1*, and *mlxlp*) in HBx,src transgenic fish with diet-induced obesity plus 40 Gy irradiation (DIO + 40 Gy) or treatment of Oligo-Fucoidan before 40 Gy irradiation (DIO + OF + 40 Gy). (**B**) The expression of lipogenesis enzyme 1-acylglycerol-3-phosphate O-acyltransferase 4 (lysophosphatidic acid acyltransferase, delta) (*agpat4*, phospholipid phosphatase 1a (plpp1a)), transcript variant X1, mRNA (*pap*), and fatty acid synthase (*fasn*) in HBx,src transgenic fish with diet-induced obesity plus 40 Gy irradiation (DIO + 40 Gy) or treatment of Oligo-Fucoidan before 40 Gy irradiation (DIO + OF + 40 Gy). (**C**). The expression of collagen, type I, alpha 1a (*col1a1*), connective tissue growth factor a (*ctgfa*), and heparanase (*hpse*) in HBx,src transgenic fish with diet-induced obesity plus 40 Gy irradiation (DIO + 40Gy) or treatment of Oligo-Fucoidan before 40 Gy irradiation (DIO + OF + 40 Gy). (**D**). The expression of cyclin E1 (*ccne1*), cyclin-dependent kinase 1 (*cdk1*), and cyclin-dependent kinase 2 (*cdk2*) in HBx, src transgenic fish with diet-induced obesity plus 40 Gy irradiation (DIO + 40 Gy) or treatment of Oligo-Fucoidan before 40 Gy irradiation (DIO + OF + 40 Gy). (**E**). The expression of *col1a1*, *ctgfa*, and *hpse* in HBx, src, p53-transgenic fish with diet-induced obesity plus 10 Gy irradiation (DIO + 10 Gy) or treatment of Oligo-Fucoidan before 10 Gy irradiation (DIO + OF + 10 Gy). (**F**). The expression of *ccne1*, *cdk1,* and *cdk2* in HBx,src,p53-transgenic fish with diet-induced obesity plus 10 Gy irradiation (DIO + 10 Gy) or treatment of Oligo-Fucoidan before 10 Gy irradiation (DIO + OF + 10 Gy). The data are presented as dot plots with a horizontal line for the mean and are repeated in triplicate. The statistical significance was calculated using Student’s *t*-test (* *p <* 0.05, ** *p <* 0.01, ns: non-significant).

**Figure 4 cancers-12-01608-f004:**
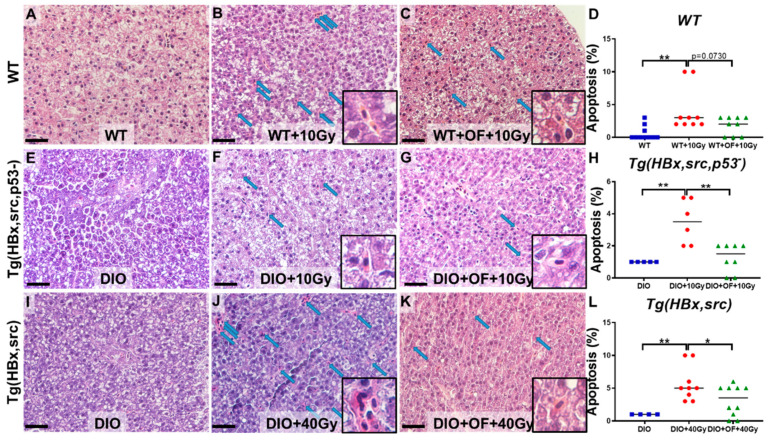
Representative images of hepatocyte apoptosis detected on hematoxylin and eosin (H&E) stain liver specimen from WT and transgenic fish with radiation without Oligo-Fucoidan or with Oligo-Fucoidan pretreatment. The images were taken at 400× magnification, and the scale shown is for 30 µm; the box area is enlarged to show the hepatocyte apoptosis, and the blue arrows pointed to the apoptotic hepatocytes. (**A–C**) The hepatocyte apoptosis in wild-type fish (WT) with 10 Gy irradiation (WT + 10 Gy) or treatment of Oligo-Fucoidan before 10 Gy irradiation (WT + OF + 10 Gy). (**D**) Statistical analysis of hepatocyte apoptosis feature of WT fish. (**E–G**) The hepatocyte apoptosis in Tg (HBx, src, p53−) transgenic fish with diet-induced obesity (DIO) with 10 Gy irradiation (DIO + 10 Gy) or treatment of Oligo-Fucoidan before 10 Gy irradiation (DIO + OF + 10 Gy). (**H**) Statistical analysis of hepatocyte apoptosis feature of Tg (HBx, src, p53−) fish. (**I–K**) The hepatocyte apoptosis in Tg (HBx, src) transgenic fish with diet-induced obesity (DIO) with 10 Gy irradiation (DIO + 40 Gy) or treatment of Oligo-Fucoidan before 40 Gy irradiation (DIO + OF + 40 Gy). (**L**) Statistical analysis of hepatocyte apoptosis feature of Tg (HBx, src) fish. The data are presented as dot plots with a horizontal line for the mean and are repeated in triplicate. The statistical significance was calculated using Student’s *t*-test (* *p <* 0.05, ** *p <* 0.01).

**Figure 5 cancers-12-01608-f005:**
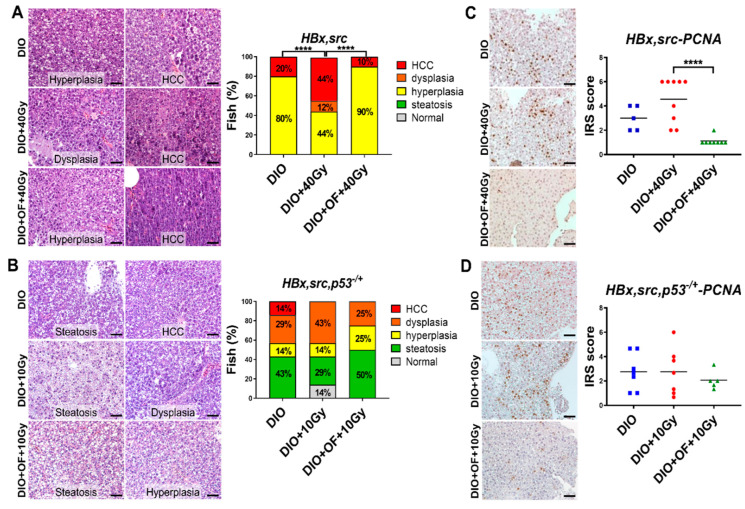
Histopathological analysis by H&E staining of hepatocytes in adult HBx, src transgenic zebrafish treated with Oligo-Fucoidan and radiation. (**A,B**) Representative images for H&E stain were shown. The images were taken at 400× magnification, and the scale shown is for 30 µm. (**A**) Statistical analysis of H&E staining of the liver sections from HBx, src transgenic fish diet-induced obesity (DIO) with 40 Gy irradiation (DIO + 40 Gy) and oral feeding Oligo-Fucoidan (DIO + OF + 40 Gy). (**B**) Statistical analysis of H&E staining of the liver sections from HBx,src,p53-transgenic fish diet-induced obesity (DIO) with 10 Gy irradiation (DIO + 10 Gy) and oral feeding Oligo-Fucoidan (DIO + OF + 10 Gy). The different colors denote the different pathological features as follows: gray: normal, green: steatosis, yellow: hyperplasia, orange: dysplasia, and red: hepatocellular carcinoma (HCC). N = 4–5 for the DIO group, and N = 9–10 for the other groups. (**C****,D**) Representative images for proliferating cell nuclear antigen (PCNA) immunohistochemistry (IHC) stain were shown. The images were taken at 400× magnification, and the scale shown is for 30 µm. (**C**) Statistical analysis of PCNA immunohistochemistry staining of the liver sections from HBx,src transgenic fish diet-induced obesity (DIO) with 40 Gy irradiation (DIO + 40 Gy) and oral feeding Oligo-Fucoidan (DIO + OF + 40 Gy). (**D**) Statistical analysis of PCNA IHC staining of the liver sections from HBx,src,p53-transgenic fish diet-induced obesity (DIO) with 10 Gy irradiation (DIO + 10 Gy) and oral feeding Oligo-Fucoidan (DIO + OF + 10 Gy). The data are presented as dot plots with a horizontal line for the mean and are repeated in triplicate. The statistical significance was calculated using Student’s *t*-test (**** *p <* 0.0001).

**Figure 6 cancers-12-01608-f006:**
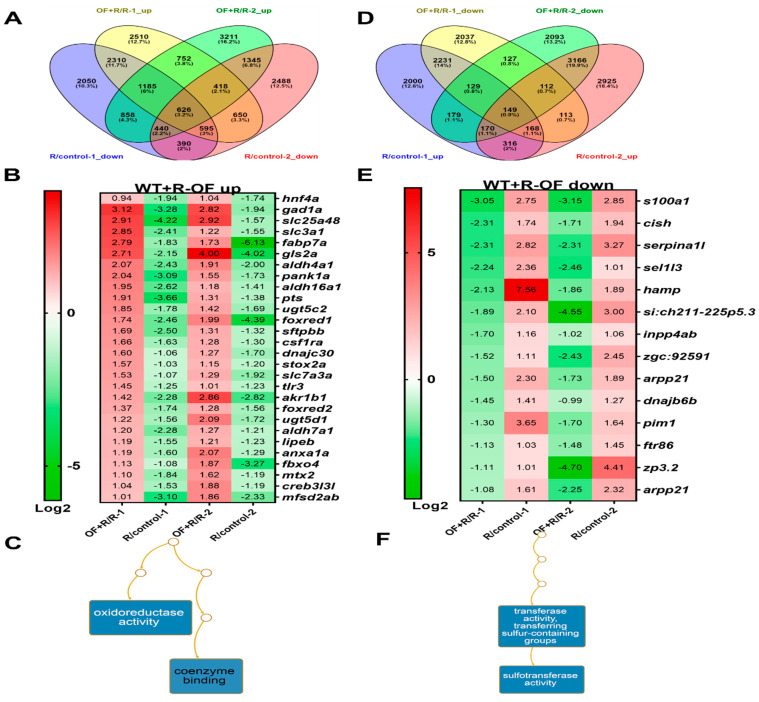
GeneTitan array analysis of the expression profile of wild-type zebrafish following different treatments. (**A**) Venn diagram of upregulation of genes in WT zebrafish irradiation and Oligo-Fucoidan pretreatment (OF + R) versus irradiation alone (R), overlapping with downregulated in irradiation (R) compare to no radiation (control). (**B**) The heatmap of genes that were upregulated in Oligo-Fucoidan pretreatment but downregulated by irradiation. First batch is 40 Gy (1), and second batch is 10 Gy (2). **(C)** Gene ontology analysis of Oligo-Fucoidan-induced genes in WT irradiation (**D**) Venn diagram of downregulation of genes in WT zebrafish irradiation and Oligo-Fucoidan pretreatment (OF + R) versus irradiation alone (R), overlapping with upregulated in irradiation (R) compared to no radiation (control). (**E**) The heatmap of genes that were downregulated in Oligo-Fucoidan pretreatment but upregulated by irradiation. First batch is 40 Gy (1), and second batch is 10 Gy (2). (**F**) Gene ontology analysis of Oligo-Fucoidan repressed genes in WT irradiation.

**Figure 7 cancers-12-01608-f007:**
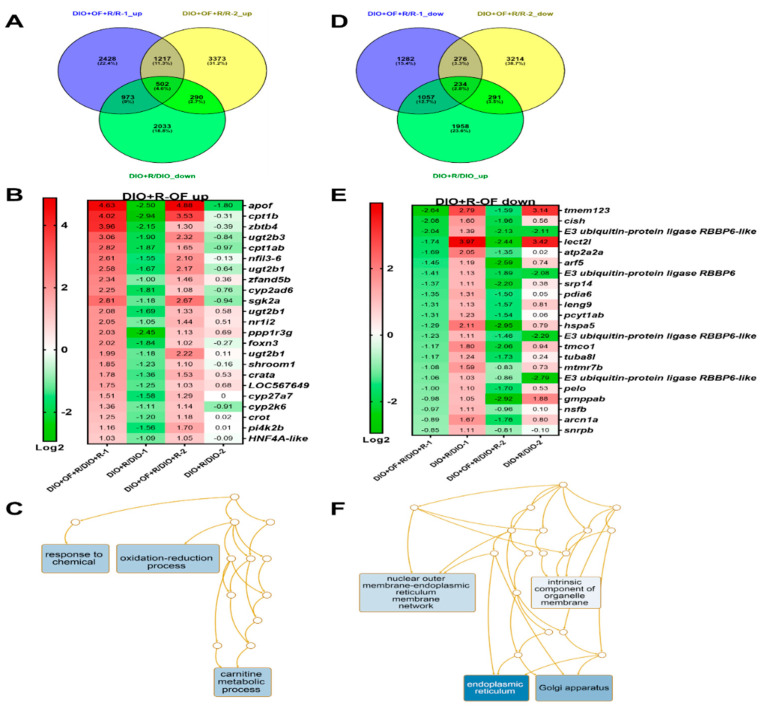
GeneTitan array analysis of the expression profile of HCC transgenic zebrafish following different treatments. (**A**) Venn diagram of upregulation of genes in HCC transgenic zebrafish diet-induced obesity with irradiation and Oligo-Fucoidan pretreatment (DIO + OF + R) compared to irradiation only (DIO + R), overlapping with downregulated in irradiation (DIO + R) compared to no radiation (DIO). (**B**) The heatmap of genes upregulated in Oligo-Fucoidan pretreatment but downregulated by irradiation. First batch is HBx, src transgenic fish with 40 Gy (1), and second batch is HBx, src, p53-transgenic fish with 10 Gy (2). (**C**) Gene ontology analysis of Oligo-Fucoidan induced genes in irradiated transgenic fish. (**D**) Venn diagram of downregulation of genes in HCC transgenic zebrafish diet-induced obesity with irradiation and Oligo-Fucoidan pretreatment (DIO + OF +R) compared to irradiation only (DIO + R), overlapping with downregulated in irradiation (DIO + R) compare to no radiation (DIO). (**E**) The heatmap of genes downregulated in Oligo-Fucoidan pretreatment but upregulated by irradiation. First batch is HBx, src transgenic fish with 40 Gy (1), and second batch is HBx, src, p53-transgenic fish with 10 Gy (2). (**F**) Gene ontology analysis of Oligo-Fucoidan repressed genes in irradiated transgenic fish.

**Figure 8 cancers-12-01608-f008:**
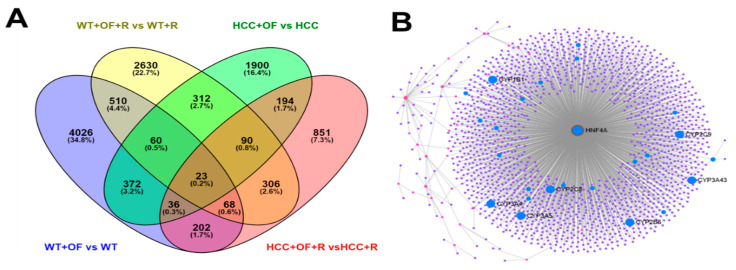
GeneTitan array analysis of expression profiles of wild-type and *HBx*, *src* transgenic zebrafish following different treatments. (**A**) Venn diagram of overlapping differentially expressed genes. (**B**) *HNF4A* is the driver gene according to NetworkAnalyst (https://www.networkanalyst.ca/).

**Figure 9 cancers-12-01608-f009:**
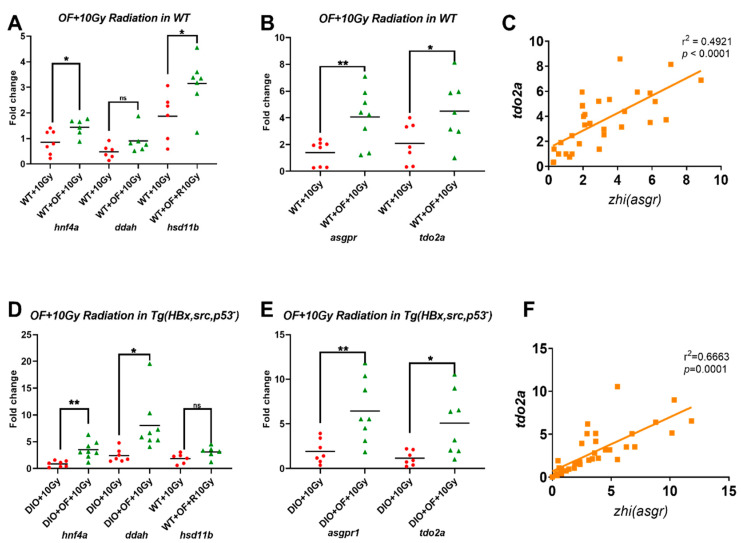
Upregulations of *hnf4a*, *ddah1*, *11b-hsd1*, *asgpr1*, and hnf4a downstream target genes: *tdo2a*, in zebrafish after irradiation. (**A**) Gene expression profiles of *hnf4a*, *ddah1*, *11b-hsd1* and (**B**) *zhi (asgpr1)*, *tdo2a* in WT zebrafish irradiation (WT + 10 Gy) and Oligo-Fucoidan pretreatment (WT + OF + R). (**C**) Positive correlations between *zhi* (*asgpr1*) and *tdo2a* were shown. (**D**) Gene expression profiles of *hnf4a*, *ddah1*, *11b-hsd1* and (**E**) *zhi (asgpr1), tdo2a* in HBx,src,p53-transgenic fish with diet-induced obesity plus 10 Gy irradiation (DIO + 10 Gy) or treatment of Oligo-Fucoidan before 10 Gy irradiation (DIO + OF + 10 Gy). (**F**) Positive correlations between *zhi (asgpr1*) *and tdo2a* were shown. Statistical significance was calculated by *t*-test (** p* ≤ 0.05, *** p* ≤ 0.01).

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
