# Peer review of "Low Molecular Weight Fucoidan Prevents Radiation-Induced Fibrosis and Secondary Tumors in a Zebrafish Model"

_cancers, 2020, doi:10.3390/cancers12061608_

Round 1
Reviewer 1 Report
The manuscript entitled “Low molecular weight Fucoidan prevents radiation induced fibrosis and secondary tumors in a zebrafish model” by Wu et al. shows that oligo-Fucoidan (OF) gavage has a radioprotective effect in wild-type zebrafish as well as in transgenic cancer liver model. Transcriptomic analyses revealed that OF enhance expression of genes involved in oxidorectuctase activity. The authors suggest that OF could prevent at least in some extend radiation-induced liver fibrosis and second malignancies.
The manuscript is clear, very well presented and written. The approach is based on an original model. The experiments are well conducted and the conclusions are supported by the experimental data. However, some points of the manuscript would require some clarification. In particular:
- I would suggest the authors to comment on the rational of the genes chosen as markers for cell death, fibrosis and proliferation.
- Lane 93-94: “The expression of these fibrosis-related genes was increased after (…)”. It is not the case for hpse. Please clarify.
- Figure 2 part C: WT+R(10Gy) should be WT+10Gy
- Lane 141: Please specify whether the transgenic fish used for the expression studies all develop hepatocellular carcinoma.
- Since the expression levels (Fig. 1, 3, S1, S2 and S3) are expressed as a fold change, it is not clear whether there is a difference between wild-type and transgenic cancer liver fish in terms of relative levels of expression for the fibrotic genes.
Author Response
Reviewer 1 Comments and Suggestions for Authors
The manuscript entitled “Low molecular weight Fucoidan prevents radiation induced fibrosis and secondary tumors in a zebrafish model” by Wu et al. shows that oligo-Fucoidan (OF) gavage has a radioprotective effect in wild-type zebrafish as well as in transgenic cancer liver model. Transcriptomic analyses revealed that OF enhance expression of genes involved in oxidorectuctase activity. The authors suggest that OF could prevent at least in some extend radiation-induced liver fibrosis and second malignancies.
The manuscript is clear, very well presented and written. The approach is based on an original model. The experiments are well conducted and the conclusions are supported by the experimental data. However, some points of the manuscript would require some clarification. In particular:
- I would suggest the authors to comment on the rational of the genes chosen as markers for cell death, fibrosis and proliferation.
Reply: Thank you very much for your constructive advice. We have added the rationale for choosing the genes as markers for cell death, fibrosis, and proliferation as following:
Lane 116-123
Activation of caspases are essential for apoptosis, including initiators (caspase-2,-8,-9,-10), effectors (caspase-3,-6,-7) and inflammatory caspases (caspase-1,-4,-5). The intrinsic signaling pathways of apoptosis is regulated by Bcl-2 family of proteins, among them, Bad (BCL2 antagonist of cell death) and Bax (BCL2 associated X protein) are pro-apoptotic proteins. Fas ligand (FasL) is initial step for extrinsic signaling pathways of apoptosis [34]. c-Jun N-terminal kinase (JNK) and p38 MAPK are key regulators of cell stress and often associated with apoptosis [35]. Therefore, we choose to examine the expression levels for caspase8, caspase3a, bad, bax, fasL, jnk-1 and p38a(MAPK) as markers for cell death.
Lane 128- 133
Type I collagen is associated with hepatic fibrosis [36], type I procollagen is combined with two pro-alpha1(I) and one pro-alpha2(I) chain which are encoded by COL1A1 and COL1A2 respectively [37]. Connective tissue growth factor (CTGF) is a fibrogenic master switch in liver fibrosis [38], Heparanase (HPSE) expression increased in the onset of liver fibrosis of CCl4-treated mice model [39]. Therefore, we choose to examine the expression levels for col1a1, ctgfa, and hpse as markers for liver fibrosis.
Lane 137- 141
The initiation of hepatocellular carcinoma depends on E-type cyclins E1 (CcnE1) and cyclin-dependent kinase 2 (Cdk2)[40], Ccne1 overexpression cause liver tumor development in mice [41], Cdk2 plays a key role in cell cycle progression in hepatocyte [42], Cyclin-dependent kinase 1 (Cdk1) is essential for cell division of liver cancer [43]. Therefore, we choose to examine the expression levels for ccne1, cdk1, and cdk2 as markers for cell proliferation.
- Lane 93-94: “The expression of these fibrosis-related genes was increased after (…)”. It is not the case for hpse. Please clarify.
Reply: Thank you very much for your suggestion. We have edited the sentence as following:
Lane 134-135
The expression of col1a1 and ctgfa was increased in the liver after RT relative to non-irradiated fish, but the increment of hpse was not significant (Fig S1B).
- Figure 2 part C: WT+R(10Gy) should be WT+10Gy
Reply: Thank you very much for your correction, we have edited the Figure 2 part C to WT+10Gy.
- Lane 141: Please specify whether the transgenic fish used for the expression studies all develop hepatocellular carcinoma.
Reply: Thank you very much for your comments, not all of the fish developed HCC, we have clarify the statement as following:
Lane 184-186
Diet-induced Obesity (DIO) accelerate HCC formation in HBx, src transgenic fish at 5 months of age when compared to normal diet, 23% of HCC under DIO versus 7% of HCC in normal diet [33].
- Since the expression levels (Fig. 1, 3, S1, S2 and S3) are expressed as a fold change, it is not clear whether there is a difference between wild-type and transgenic cancer liver fish in terms of relative levels of expression for the fibrotic genes.
Reply: Thank you very much for your comments, all of the fold change were compared to wild-type fish normal diet, so there is a significant difference between wild-type and transgenic fish with liver cancer in terms of the expression for fibrotic genes and proliferation markers. But, we focus on radio-protection effect, so we did not emphasize this.

Reviewer 2 Report
The report suggested the oral intake of Fucoidan may protect the fibrosis and development of secondary tumors after radiation treatment. The paper underwent several set of genes for gene expression. However, the data is not clearly described and the discussion is lack of content. I have the following specific comments:
Major:
L68: Authors introduced the Fucoidan with several references. Please specific which models were used. The data seems come from the mammalian model or cell line models. Could authors find some references in fishes? Furthermore, Fucoidan is the keyword in this study. Authors should include more background of that.
L74: Have zebrafish be used for RT in researches? References should be given.
Figure 1. Did the authors check the critical p53 expression?
Furthermore, one of the concern is the DNA damage caused by RT. Did author check anything on that? Comet assay on wild-type? And wild-type+FO?
Moreover, it will be important to check the protein expression instead of the mRNA expression for those caspase and cell death proteins.
L178: I wonder if the fish can survive after RT, and if so, did the authors observe the development of secondary tumor afterwards? This is important to claim the FO could prevent the development of secondary tumor.
L185: Please provide the staining image for reference.
After bioinformatics analysis, authors identified various potential pathway The RT-PCR validation did not provide any insight for the data. To increase the merit of the paper, authors should include some functional experiments.
L266, what if those receptors were knocked-down/ KO. Will FO treatment still affect the similar set of genes?
The discussion is far too short, and mainly repeating the data description. What are the significant and the merits of the data? Authors obtained the genome-wide data, could they discuss them? I have to say that I cannot get any credits from the discussion.
Minor:
Any references for the RT treatment in zebrafish? L328
L342/L354: the molecular weight varies a lot (500-1500Da). I wonder if the size affects the binding efficiency and how the authors could ensure the experimental fish ate “similar amount” as others? Any
Primer sequences can be showed as supplementary.
Author Response
Reviewer 2 Comments and Suggestions for Authors
The report suggested the oral intake of Fucoidan may protect the fibrosis and development of secondary tumors after radiation treatment. The paper underwent several set of genes for gene expression. However, the data is not clearly described and the discussion is lack of content. I have the following specific comments:
Major:
L68: Authors introduced the Fucoidan with several references. Please specific which models were used. The data seems come from the mammalian model or cell line models. Could authors find some references in fishes? Furthermore, Fucoidan is the keyword in this study. Authors should include more background of that.
Reply: Thank you very much for your constructive advice. We have found some references in fishes and rat for radiation and fucoidan experiment, we also include more background of fucoidan as following:
Lane 97-104
Zebrafish embryos was used as a model to screen radiation modifiers [30], ionizing radiation ranging from 10 to 40 Gy caused time- and dose-dependent perturbations and lethality. Adult zebrafish irradiated with 20 Gy can impact the outcome of hematopoietic cell transplant was reported [31]. However, there is no literature for using adult zebrafish for radio-protection. Therefore, the dosage was determined based on previous study in rat that fucoidan from Laminaria japonica at 300 mg/kg body weight per day has no adverse effect [52]. Herein, zebrafish adult fish were feeding with oligo-fucoidan (OF) by oral gavage, the dose was 300 mg/kg (0.051 mg/fish).
L74: Have zebrafish be used for RT in researches? References should be given.
Reply: Thank you very much for your suggestion. The zebrafish for radiation therapy was performed using zebrafish embryos, the references were given in the revised manuscript as in lane 97-104.
Figure 1. Did the authors check the critical p53 expression?
Reply: Thank you very much for your comments. We did not check p53 expression in wild-type fish, and we used p53 mutant fish as cancer model, so the p53 expression was not measured.
Furthermore, one of the concern is the DNA damage caused by RT. Did author check anything on that? Comet assay on wild-type? And wild-type+FO?
Reply: Thank you very much for your question. We did not check radiotherapy induced DNA damage by comet assay, because previous studies has lots of data on it. In this study, we are focusing on radiation cause fibrosis and secondary tumor in adult zebrafish model.
Moreover, it will be important to check the protein expression instead of the mRNA expression for those caspase and cell death proteins.
Reply: Thank you very much for your critical comments. We did not check protein expression because we are using zebrafish model, lots of the antibodies were generated to react with human or mouse protein, not so much for zebrafish.
L178: I wonder if the fish can survive after RT, and if so, did the authors observe the development of secondary tumor afterwards? This is important to claim the FO could prevent the development of secondary tumor.
Reply: Thank you very much for your critical comments. The fish survive after RT, and we sacrificed the fish one week after radiation. We observed the OF prevent the development of secondary tumor based on our HE stain result, where 40Gy radiation increased the HCC formation, and OF pretreatment decreased the HCC percentage in the fish as shown in Figure 4.
L185: Please provide the staining image for reference.
Reply: Thank you very much for your suggestion. We have added representative images for H&E stain and PCNA IHC staining in the Figure 4.
After bioinformatics analysis, authors identified various potential pathway The RT-PCR validation did not provide any insight for the data. To increase the merit of the paper, authors should include some functional experiments.
Reply: Thank you very much for your comments. Total 626 candidate genes were downregulated by irradiation and reverted by oligo-fucoidan treatment (Fig 5A, B). Using gene ontology analysis via WebGestalt [39], we found genes involved in oxidoreductase activity were enriched (Fig 5C). All the differentially expressed genes were uploaded to the NetworkAnalyst software (https://www.networkanalyst.ca/), hnf4a was the hub for most of the differentially expressed genes. Therefore, hnf4a was determined to be a driver gene. Therefore, we examined how fucoidan-mediated activation of hnf4a and its downstream targets related to fibrosis and cancer formation using RT-PCR validation.
L266, what if those receptors were knocked-down/ KO. Will FO treatment still affect the similar set of genes?
Reply: Thank you for the constructive suggestion. This manuscript is focusing on radio protection effect of Oligo-fucoidan, so we did not show the knockdown the receptor or inhibit the STAT3 effect. In another manuscript, we have found OF induced increased the cell viability of normal hepatocyte (Clone 9), knockdown Asgpr1 in normal hepatocyte eliminated OF’s effect (below Fig. 1) and genes’ expression (not shown). We also used STAT3 inhibitor to treat the normal hepatocyte (Clone 9), and found the STAT3 inhibitor decreased the viability and the Cyp3a4 activity of normal liver cells (below Fig. 2A, B). Those result indicated Asgpr and Stat3 is involved in the hepatocyte viability. We have used STAT3 inhibitor to treat the normal hepatocyte (Clone 9), and confirmed the expression of Hnf4a is regulated by Stat3 (below Fig. 2C).
Fig. 1
Fig. 2
The discussion is far too short, and mainly repeating the data description. What are the significant and the merits of the data? Authors obtained the genome-wide data, could they discuss them? I have to say that I cannot get any credits from the discussion.
Reply: Thank you very much for your critical comments. We have added more discussion on the bioinformatic results and the fucoidan radioprotection effect, we also discuss other radio protectants and explain why oligo-fucoidan is better than others.
Minor:
Any references for the RT treatment in zebrafish? L328
Reply: Thank you very much for your advice. We have found some references RT treatment in zebrafish as following:
Lane 97-101
Zebrafish embryos was used as a model to screen radiation modifiers [30], ionizing radiation ranging from 10 to 40 Gy caused time- and dose-dependent perturbations and lethality. Adult zebrafish irradiated with 20 Gy can impact the outcome of hematopoietic cell transplant was reported [31]. However, there is no literature for using adult zebrafish for radio-protection.
L342/L354: the molecular weight varies a lot (500-1500Da). I wonder if the size affects the binding efficiency and how the authors could ensure the experimental fish ate “similar amount” as others? Any
Reply: Thank you very much for your comments, in this study, we used low molecular weight fucoidan has an average molecular weight of 800 Da in the range from 500 to 1500 Da. The low molecular weight fucoidan has more biological actions than native fucoidan. We ensure the experimental fish ate “similar amount” as others by injected 5 μl of fucoidan solution into the intestinal tract. So we clarify the methods as below:
Lane 101-104
Therefore, the dosage was determined based on previous study in rat that fucoidan from Laminaria japonica at 300 mg/kg body weight per day has no adverse effect [52]. Herein, zebrafish adult fish were feeding with oligo-fucoidan (OF) by oral gavage, the dose was 300 mg/kg (0.051 mg/fish).
Primer sequences can be showed as supplementary.
Reply: Thank you very much for your comments, the primer sequences has moved to supplementary Table S1.

Reviewer 3 Report
This study shows that gavage of wild zebrafish with oligofucoidan reduces the expression of apoptotic genes and prevents radiation-induced fibrosis.
In transgenic zebrafish highly associated with liver cancer formation, oligofucoidan gavage decreases the expression of lipogenic factors and lipogenic enzymes as well as fibrosis in the presence of a fatty diet and even hepatocarcinogenesis.
On these animals geneontology analysis revealed that OF pretreatement rescue the expression of genes which have been modulated by irradiation thus promoting the prevention of cancer and the formation of second malignancies.
The choice of the quantities of rays and at one time is not sufficiently explained in relation to the clinical rationale.
The discussion could be further improved, in particular with the addition of some references: only 12 references are cited in this section.
With regard to the literature, it is not exact to write that “this is the first article to demonstrate an effective agent in preventing radiation-induced secondary cancers”. It would be indeed appropriate to clarify why oligo-fucoidan seems more promising than other compounds cited in the past or current literature (e.g. amifostine, nicaraven, zoledronate tangeretin...).
Just one citation in the discussion referring to the particular structure of fucoidan may seem insufficient all the more than it is not described elsewhere.
Oral gavage is done before radiation, is it done in the same quantity in wild and transgenic animals?
In figure S1, problem with the legend of D idem in figure 1,D
There is no match between the points in Figure 1A and Figure S1A for column 40Gy for the bad gene.
In Figure 1, on the other hand, it is very surprising to have exactly the same point profile for the baxa and caspase3egenes under both 40Gy and OF+40Gy conditions.
In figure 2 there is an error in the legend of C. It is "before 10Gy".
Pictures associated with the PCNA proliferation markers would be welcome and this method needs to be described in the material and methods.
In figure 4, the legends refer only to 40Gy but on figures B and D it is indicated 10Gy.
In the results of figures 5 and 6 it is not clarified whether the radiation control is 40Gy or 10Gy without OF.
On line 272 the authors refer to another project. In case where the data are published it would be preferable to said publication or otherwise indicate that the data are preliminary.
Author Response
Reviewer 3 Comments and Suggestions for Authors
This study shows that gavage of wild zebrafish with oligofucoidan reduces the expression of apoptotic genes and prevents radiation-induced fibrosis.
In transgenic zebrafish highly associated with liver cancer formation, oligofucoidan gavage decreases the expression of lipogenic factors and lipogenic enzymes as well as fibrosis in the presence of a fatty diet and even hepatocarcinogenesis.
On these animals geneontology analysis revealed that OF pretreatement rescue the expression of genes which have been modulated by irradiation thus promoting the prevention of cancer and the formation of second malignancies.
The choice of the quantities of rays and at one time is not sufficiently explained in relation to the clinical rationale.
Reply: Thank you very much for your comments. Using whole genomic microarray, we compared two batches of wild-type fish radiation with or without OF pretreatment, we identified 626 candidate genes were downregulated by irradiation, and upregulated by oligo-fucoidan pretreatment before radiation (Fig 5A, B), and those genes were enriched in oxidoreductase activity (Fig 5C). We also identified 149 candidate genes were upregulated by irradiation, and down-regulated by oligo-fucoidan pretreatment before radiation (Fig 5D, E), and those genes enriched in transferase activity (Fig 5F). The cut-off were at least two-fold with a p-value less than 0.05.
We compared two batches of transgenic fish radiation with or without OF pretreatment, we identified 502 genes were down-regulated by 40 Gy irradiation in HBx,src transgenic fish with DIO, and upregulated by oligo-fucoidan pretreatment before radiation in both batches, and those genes are involved in oxidoreductase activity (Fig 6A-C). We also identified 234 genes were up-regulated by 40Gy irradiation in HBx,src transgenic fish with DIO, and down-regulated by oligo-fucoidan pretreatment before radiation in both batches, and those genes are involved OF also decreased the expression of genes involved in nuclear outer membrane-endoplasmic reticulum membrane network and non-homologous end-Joining (NHEJ) in HCC transgenic fish (Fig 6D-F).
Using whole genomic microarray to screen the genes are down (or-up) regulated by irradiation and reversed by oligo-fucoidan pretreatment before irradiation is highly related to the clinical setting, and we identified genes are involved in radiation and reduced by oligo-fucoidan pretreatment proving the efficacy of oligo-fucoidan on radioprotection.
The discussion could be further improved, in particular with the addition of some references: only 12 references are cited in this section.
Reply: Thank you very much for your constructive advice. We have further improved our discussion, and cited many updated references on bioinformatics and the fucoidan radioprotection effect.
With regard to the literature, it is not exact to write that “this is the first article to demonstrate an effective agent in preventing radiation-induced secondary cancers”. It would be indeed appropriate to clarify why oligo-fucoidan seems more promising than other compounds cited in the past or current literature (e.g. amifostine, nicaraven, zoledronate tangeretin...).
Reply: Thank you very much for your comments. We compared oligo-fucoidan with other compounds (e.g. amifostine, nicaraven, zoledronate tangeretin...) for preventing radiation-induced secondary cancers.
Just one citation in the discussion referring to the particular structure of fucoidan may seem insufficient all the more than it is not described elsewhere.
We have refer more citations in the discussion regarding to the particular structure of fucoidan and its effect.
Oral gavage is done before radiation, is it done in the same quantity in wild and transgenic animals?
Reply: Thank you very much for your comments. Oral gavaging was conducted as described previously (J Vis Exp. 2013; (78): 50691), the fish were anesthetized using MS-222 solution (150 mg/L), and 5 μl of solution was injected slowly into the intestinal tract of adult fish using flexible tubing. We have rewritten the Section 4.4 (material and methods), provided the method of oral gavage and cite this paper.
The dosage was determined based on previous study in rat that fucoidan from Laminaria japonica at 300 mg/kg body weight per day has no adverse effect [52]. Herein, zebrafish adult fish were used to examine the radio-protective effects of oligo-fucoidan (OF) by feeding the 300 mg/kg dosage (0.051 mg/fish) using oral gavage.
We have found some references in fishes and rat for radiation and fucoidan experiment, we also include more background of fucoidan as following:
Lane 97-104
Zebrafish embryos was used as a model to screen radiation modifiers [30], ionizing radiation ranging from 10 to 40 Gy caused time- and dose-dependent perturbations and lethality. Adult zebrafish irradiated with 20 Gy can impact the outcome of hematopoietic cell transplant was reported [31]. However, there is no literature for using adult zebrafish for radio-protection. Therefore, the dosage was determined based on previous study in rat that fucoidan from Laminaria japonica at 300 mg/kg body weight per day has no adverse effect [52]. Herein, zebrafish adult fish were feeding with oligo-fucoidan (OF) by oral gavage, the dose was 300 mg/kg (0.051 mg/fish).
In figure S1, problem with the legend of D idem in figure 1,D
Reply: Thank you very much for your advice. We have corrected the figure legend for the Figure 1D and Figure S1D.
Figure 1D _Lane 168
(D) The expression of col1a1, ctgfa and hpse in wild-type fish 10 Gy irradiation (10 Gy)
Figure S1D _Lane 19
(D) The expression of col1a1, ctgfa and hpse in wild-type fish 10 Gy irradiation (10 Gy)
There is no match between the points in Figure 1A and Figure S1A for column 40Gy for the bad gene.
Reply: Thank you very much for the carefully noticed. After we checked the data, the Figure 1A_40Gy for the bad gene contains two additional unrelated points, so we corrected this mistake and replay the Figure 1 with new 1A.
Fig1 FigS1
In Figure 1, on the other hand, it is very surprising to have exactly the same point profile for the baxa and caspase3a genes under both 40Gy and OF+40Gy conditions.
Reply: Thank you very much for the carefully checked. After we checked the data, we found the caspase3a data was misplaced with baxa’s data, so we corrected this mistake and replaced the new Figure 1 (as shown above).
In figure 2 there is an error in the legend of C. It is "before 10Gy".
Reply: Thank you very much for the noticed. We have corrected the error in the figure legend 1C.
Lane 229: before 10 Gy irradiation (WT+OF+10Gy).
Pictures associated with the PCNA proliferation markers would be welcome and this method needs to be described in the material and methods.
Reply: Thank you very much for your suggestion. We have added representative images for H&E stain and PCNA IHC staining in the Figure 4. The PCNA IHC staining method has been added to the Section 4.9 (material and methods).
In figure 4, the legends refer only to 40Gy but on figures B and D it is indicated 10Gy.
Reply: Thank you very much for the noticed. We have corrected the error in the figure legend 4B and D to 40Gy.
Lane 307
10Gy irradiation (DIO+10Gy) and oral feeding OF (DIO+OF+10Gy).
Lane 315
10Gy irradiation (DIO+10Gy) and oral feeding OF (DIO+OF+10Gy).
In the results of figures 5 and 6 it is not clarified whether the radiation control is 40Gy or 10Gy without OF.
Reply: Thank you very much for your comments. We have clarified as following:
Lane 338-342
We compared two batches of wild-type fish radiation (1st batch-40Gy and 2nd batch- 10Gy) with or without OF pretreatment, we identified 626 candidate genes were downregulated by irradiation (R/control), and upregulated by oligo-fucoidan pretreatment before radiation (OF+R/R) at least two-fold with a p-value less than 0.05 (Fig 5A, B).
Lane 350-352
We also identified 149 candidate genes were upregulated by irradiation (R/control), and down-regulated by oligo-fucoidan pretreatment before radiation (OF+R/R) at least two-fold with a p-value less than 0.05 (Fig 5D, E), and those genes enriched in transferase activity (Fig 5F).
Lane 386-391
We compared two batches of transgenic fish radiation (1st batch-HBx,src fish diet-induced obesity (DIO) with 40Gy and 2nd batch- HBx,src,p53- transgenic fish DIO with 10Gy) with or without OF pretreatment, we identified 502 genes were down-regulated by 40 Gy irradiation in HBx,src transgenic fish with DIO (DIO+R/DIO), and upregulated by oligo-fucoidan pretreatment before radiation in both batches (DIO+OF+R/R) at least two-fold with a p-value less than 0.05, and those genes are involved in oxidoreductase activity (Fig 6A-C).
Lane 392-397
We also identified 234 genes were up-regulated by 40Gy irradiation in HBx,src transgenic fish with DIO (DIO+R/DIO), and upregulated by oligo-fucoidan pretreatment before radiation in both batches (DIO+OF+R/R) at least two-fold with a p-value less than 0.05, and those genes are involved OF also decreased the expression of genes involved in nuclear outer membrane-endoplasmic reticulum membrane network and non-homologous end-Joining (NHEJ) in HCC transgenic fish (Fig 6D-F). Rescued of those genes can prevent liver cancer formation.
On line 272 the authors refer to another project. In case where the data are published it would be preferable to said publication or otherwise indicate that the data are preliminary.
Reply: Thank you very much for your comments, actually the paper has been processing for published, but I would change the statement as “our preliminary data”
Lane 441-443
Our preliminary data has demonstrated that OF binds to ASGPR1/2 via in-vitro and in-vivo competition assay, and increases the expression of HNF4A through the JAK2/STAT3 axis in hepatocyte.

Round 2
Reviewer 2 Report
The revised MS has been improved. Regarding the new version and response from the auhtors, I have some minor but essential comments:
1) I would like to stick on the cell death issue that auhtors mentioned, regarding the figure 1/2. As a reader, I would like to know how the RT could induce the cell death to what extends. Thus, cell death imaging in the liver tissue, or information showing the positive signals are close to the fiber development will provide a solid background data for the study. Regarding the PCR, i wonder authors have tested other cell cycle genes as references Furthermore, I would like to ask about the cas9 as well.
2) I still believe that the authors could discuss more on their array data (from 6B and E). For example , authros try to talk about the hnf4, but did not provide enough information/ link on the topics. Or if the group is still working on soem mechanism, I suggest authors to pick "other target" for a more insightful discussion.
Minor: L460
hnf4 is the hub and driver gene...I feel a little bit misleading. Does authors provide solid data on the "driver gene".
Author Response
Reviewer 2_Comments and Suggestions for Authors
The revised MS has been improved. Regarding the new version and response from the auhtors, I have some minor but essential comments:
1) I would like to stick on the cell death issue that auhtors mentioned, regarding the figure 1/2. As a reader, I would like to know how the RT could induce the cell death to what extends. Thus, cell death imaging in the liver tissue, or information showing the positive signals are close to the fiber development will provide a solid background data for the study. Regarding the PCR, i wonder authors have tested other cell cycle genes as references Furthermore, I would like to ask about the cas9 as well.
Reply: Thank you very much for your comments. RT induces cell death was mediated through double-stranded DNA sensor AIM2 which senses radiation-induced DNA damage in the nucleus, then activate inflammation and cell death (Kuwahara et al., 2018). RT induces cell death depends on DNA repair capacity and the microenvironment up top of other factors (Sia et al., 2020). We followed a paper to detect the hepatocyte apoptosis (Feldstein and Gores, 2005), and examined hepatocyte apoptosis from H&E stain liver specimens to verify the apoptosis changes following OF treatment in transgenic fish. We found in our zebrafish model, radiation indeed induced apoptosis in the liver tissue from HE stain, and OF-fed zebrafish significantly reduced the hepatocyte apoptosis. Since the fibrosis was revealed from Sirius stain, we could not know whether the positive signals are close to the fiber development. Thus, we provided new figure (Fig4) to demonstrate our findings.
We have add the results in our revised manuscript (lane 288-310)
2.3. Oligo-fucoidan Pre-treatment Decreases the Radiation Induced Hepatocyte Apoptosis in Adult Zebrafish
RT induces cell death was mediated through double-stranded DNA sensor AIM2 which senses radiation-induced DNA damage in the nucleus, then activate inflammation and cell death [53]. RT induces cell death depends on DNA repair capacity and the microenvironment up top of other factors [54]. Hepatocellular apoptosis may trigger fibrosis and tumor formation [55], we then detected hepatocyte apoptosis from H&E stain liver specimens to verify the apoptosis changes following OF treatment in transgenic fish. We found in our zebrafish model, radiation indeed induced apoptosis in the liver tissue from HE stain, and OF-fed zebrafish significantly reduced the hepatocyte apoptosis (Fig 4).
Figure 4. Representative images of hepatocyte apoptosis detected on H&E stain liver specimen from WT and transgenic fish radiation without Oligo-Fucoidan or with Oligo-Fucoidan pretreatment. The images were taken at 400X magnification and scale shown is for 30 μm, the box area were enlarged to show the hepatocyte apoptosis. (A-C) The hepatocyte apoptosis in wild-type fish (WT) with 10 Gy irradiation (WT+10 Gy) or treatment of Oligo-Fucoidan before 10 Gy irradiation (WT+OF+10Gy). (D) Statistical analysis of hepatocyte apoptosis feature of WT fish. (E-G) The hepatocyte apoptosis in Tg(HBx,src, p53-) transgenic fish diet induced obesity (DIO) with 10 Gy irradiation (DIO+10 Gy) or treatment of Oligo-Fucoidan before 10 Gy irradiation (DIO+OF+10Gy). (H) Statistical analysis of hepatocyte apoptosis feature of Tg(HBx,src, p53-) fish. (I-K) The hepatocyte apoptosis in Tg(HBx,src) transgenic fish diet induced obesity (DIO) with 10 Gy irradiation (DIO+40 Gy) or treatment of Oligo-Fucoidan before 40 Gy irradiation (DIO+OF+40Gy). (L) Statistical analysis of hepatocyte apoptosis feature of Tg(HBx,src) fish. The data are presented as dot plots with a horizontal line for the mean and are repeated in triplicate. The statistical significance was calculated using Student’s t-test (*P < 0.05, **P < 0.01).
As for the cell cycle genes, we always use ccne1, cdk1 and cdk2 for cell cycle/proliferation markers in our previous papers on zebrafish HCC models (Chou et al., 2019; Lu et al., 2014; Lu et al., 2013; Sie et al., 2020; Su et al., 2019; Tu et al., 2017; Yang et al., 2019). The rationale is that the initiation of hepatocellular carcinoma depends on E-type cyclins E1 (CcnE1) and cyclin-dependent kinase 2 (Cdk2)(Sonntag et al., 2018), Ccne1 overexpression cause liver tumor development in mice (Aziz et al., 2019), Cdk2 plays a key role in cell cycle progression in hepatocyte (Hanse et al., 2009), Cyclin-dependent kinase 1 (Cdk1) is essential for cell division of liver cancer (Diril et al., 2012). Therefore, we choose to examine the expression levels for ccne1, cdk1, and cdk2 as markers for cell proliferation.
2) I still believe that the authors could discuss more on their array data (from 6B and E). For example , authros try to talk about the hnf4, but did not provide enough information/ link on the topics. Or if the group is still working on soem mechanism, I suggest authors to pick "other target" for a more insightful discussion.
Reply: Thank you very much for your suggestion. We have perform the network analysis on fucoidan induced 23 overlapping differentially expressed genes from various insults (radiation, oncogenic factors, and diet induced obesity). We found HNF4A is the driver gene. Thus, we believe HNF4A plays a central role for different stress. We therefore edited the revised manuscript by adding following statement to emphasize the importance of hnf4a and how we found it.
The following statement has been added to our manuscript (lane 500-508):
In order to explore fucoidan-mediated gene expression in hepatocytes, the whole-genome expression profiles of four groups of fish (wild-type (WT), WT with irradiation, HCC, and HCC with irradiation) treated with oligo-fucoidan were analyzed by GeneTitan array, and 23 candidate genes displaying statistically significant differential expression were filtered out (Fig 8A), and HNF4A was determined to be a driver gene after analysis by NetworkAnalyst (https://www.networkanalyst.ca/) (Fig 8B).
Figure 8. GeneTitan array analysis of expression profiles of wild-type and HBx,src transgenic zebrafish following different treatments. (A) Venn diagram of overlapping differentially expressed genes. (B) HNF4A is the driver gene according to NetworkAnalyst (thttps://www.networkanalyst.ca/).
Minor: L460
hnf4 is the hub and driver gene...I feel a little bit misleading. Does authors provide solid data on the "driver gene".
Reply: Thank you very much for your advice. As describe above, we used those differentially expressed genes in the NetworkAnalyst software, and found out Hnf4a interact with other genes, so Hnf4a is the hub of the networks, and thus Hnf4a is the driver gene and regulates other differentially expressed genes.
References:
Aziz, K., Limzerwala, J.F., Sturmlechner, I., Hurley, E., Zhang, C., Jeganathan, K.B., Nelson, G., Bronk, S., Fierro Velasco, R.O., van Deursen, E.J., et al. (2019). Ccne1 Overexpression Causes Chromosome Instability in Liver Cells and Liver Tumor Development in Mice. Gastroenterology 157, 210-226 e212.
Chou, Y.T., Chen, L.Y., Tsai, S.L., Tu, H.C., Lu, J.W., Ciou, S.C., Wang, H.D., and Yuh, C.H. (2019). Ribose-5-phosphate isomerase A overexpression promotes liver cancer development in transgenic zebrafish via activation of ERK and beta-catenin pathways. Carcinogenesis 40, 461-473.
Diril, M.K., Ratnacaram, C.K., Padmakumar, V.C., Du, T., Wasser, M., Coppola, V., Tessarollo, L., and Kaldis, P. (2012). Cyclin-dependent kinase 1 (Cdk1) is essential for cell division and suppression of DNA re-replication but not for liver regeneration. Proc Natl Acad Sci U S A 109, 3826-3831.
Feldstein, A.E., and Gores, G.J. (2005). Apoptosis in alcoholic and nonalcoholic steatohepatitis. Front Biosci 10, 3093-3099.
Hanse, E.A., Nelsen, C.J., Goggin, M.M., Anttila, C.K., Mullany, L.K., Berthet, C., Kaldis, P., Crary, G.S., Kuriyama, R., and Albrecht, J.H. (2009). Cdk2 plays a critical role in hepatocyte cell cycle progression and survival in the setting of cyclin D1 expression in vivo. Cell Cycle 8, 2802-2809.
Kuwahara, Y., Tomita, K., Urushihara, Y., Sato, T., Kurimasa, A., and Fukumoto, M. (2018). Association between radiation-induced cell death and clinically relevant radioresistance. Histochem Cell Biol 150, 649-659.
Lu, J.W., Liao, C.Y., Yang, W.Y., Lin, Y.M., Jin, S.L., Wang, H.D., and Yuh, C.H. (2014). Overexpression of endothelin 1 triggers hepatocarcinogenesis in zebrafish and promotes cell proliferation and migration through the AKT pathway. PLoS One 9, e85318.
Lu, J.W., Yang, W.Y., Tsai, S.M., Lin, Y.M., Chang, P.H., Chen, J.R., Wang, H.D., Wu, J.L., Jin, S.L., and Yuh, C.H. (2013). Liver-specific expressions of HBx and src in the p53 mutant trigger hepatocarcinogenesis in zebrafish. PLoS One 8, e76951.
Sia, J., Szmyd, R., Hau, E., and Gee, H.E. (2020). Molecular Mechanisms of Radiation-Induced Cancer Cell Death: A Primer. Front Cell Dev Biol 8, 41.
Sie, Z.L., Li, R.Y., Sampurna, B.P., Hsu, P.J., Liu, S.C., Wang, H.D., Huang, C.L., and Yuh, C.H. (2020). WNK1 Kinase Stimulates Angiogenesis to Promote Tumor Growth and Metastasis. Cancers (Basel) 12.
Sonntag, R., Giebeler, N., Nevzorova, Y.A., Bangen, J.M., Fahrenkamp, D., Lambertz, D., Haas, U., Hu, W., Gassler, N., Cubero, F.J., et al. (2018). Cyclin E1 and cyclin-dependent kinase 2 are critical for initiation, but not for progression of hepatocellular carcinoma. Proc Natl Acad Sci U S A 115, 9282-9287.
Su, Z.L., Su, C.W., Huang, Y.L., Yang, W.Y., Sampurna, B.P., Ouchi, T., Lee, K.L., Wu, C.S., Wang, H.D., and Yuh, C.H. (2019). A Novel AURKA Mutant-Induced Early-Onset Severe Hepatocarcinogenesis Greater than Wild-Type via Activating Different Pathways in Zebrafish. Cancers (Basel) 11.
Tu, H.C., Hsiao, Y.C., Yang, W.Y., Tsai, S.L., Lin, H.K., Liao, C.Y., Lu, J.W., Chou, Y.T., Wang, H.D., and Yuh, C.H. (2017). Up-regulation of golgi alpha-mannosidase IA and down-regulation of golgi alpha-mannosidase IC activates unfolded protein response during hepatocarcinogenesis. Hepatol Commun 1, 230-247.
Yang, W.Y., Rao, P.S., Luo, Y.C., Lin, H.K., Huang, S.H., Yang, J.M., and Yuh, C.H. (2019). Omics-based Investigation of Diet-induced Obesity Synergized with HBx, Src, and p53 Mutation Accelerating Hepatocarcinogenesis in Zebrafish Model. Cancers (Basel) 11.
